# Coordinated neuron-glia regeneration through Notch signaling in planarians

M. Lucila Scimone[1,2], Bryanna Isela-Inez Canales[2,3], Patrick Aoude[2,4], Kutay D. Atabay[2], Peter W. Reddien[1,2,3]*

1 Howard Hughes Medical Institute, Massachusetts Institute of Technology, Cambridge, Massachusetts, United States of America, 2 Whitehead Institute for Biomedical Research, Cambridge, Massachusetts, United States of America, 3 Department of Biology, Massachusetts Institute of Technology, Cambridge, Massachusetts, United States of America, 4 Department of Computational and Systems Biology, Massachusetts Institute of Technology, Cambridge, Massachusetts, United States of America

* reddien@wi.mit.edu

**Data Availability Statement:** The datasets generated during and/or analyzed during the current study are available in the NCBI Sequence Read Archive (SRA) under the accession numbers: SRA: PRJNA1067154 and PRJNA1087310.

## Abstract

Some animals can regenerate large missing regions of their nervous system, requiring mechanisms to restore the pattern, numbers, and wiring of diverse neuron classes. Because injuries are unpredictable, regeneration must be accomplished from an unlimited number of starting points. Coordinated regeneration of neuron-glia architecture is thus a major challenge and remains poorly understood. In planarians, neurons and glia are regenerated from distinct progenitors. We found that planarians first regenerate neurons expressing a Delta-encoding gene, *delta-2*, at key positions in the central and peripheral nervous systems. Planarian glia are specified later from dispersed Notch-1-expressing mesoderm-like phagocytic progenitors. Inhibition of *delta-2* or *notch-1* severely reduced glia in planarians, but did not affect the specification of other phagocytic cell types. Loss of several *delta-2*-expressing neuron classes prevented differentiation of the glia associated with them, whereas transplantation of *delta-2*-expressing photoreceptor neurons was sufficient for glia formation at an ectopic location. Our results suggest a model in which patterned *delta-2*-expressing neurons instruct phagocytic progenitors to locally differentiate into glia, presenting a mechanism for coordinated regeneration of numbers and pattern of cell types.

## Author summary

Planarians are able to regenerate every part of their body following injury including their nervous system. Regeneration of the nervous system faces the challenge of restoring the correct pattern and number of both neurons and glia. Neurons and glia in planarians are specified from different progenitor populations, presenting the challenge of coordinating neuron-glia number and pattern in regeneration. We found that neurons expressing a Delta ligand regenerate in a patterned manner first, with dispersed Notch-1+ glia progenitors interacting with these neurons to regenerate glia. We propose a model for coordinated regeneration where interaction between Notch and Delta triggers the regeneration of neurons and glia in a patterned manner at specific locations.

**Funding:** This work was support by the National Institutes of Health R35 GM145345 to P.W.R. This work was also supported by the Howard Hughes Medical Institute and the Eleanor Schwartz Charitable Foundation. The funders had no role in the study design, data collection and analysis, decision to publish, or preparation of the manuscript. P.W.R. and M.L.S. received salary from the Howard Hughes Medical Institute.

**Competing interests:** The authors have declared that no competing interests exist.

## Introduction

In animal regeneration certain cell types might need to form in correct relative numbers and positions, a problem referred to here as coordinated regeneration. For example, neurons and glia for certain contexts need to be regenerated in a coordinated architecture, even if they arise from different progenitors. We address this problem of coordinated regeneration of neurons and glia in planarians.

Planarians are freshwater flatworms that belong to the Spiralian superphylum [1] and are capable of whole-body regeneration and constant tissue turnover. These processes involve a proliferating population of stem cells called neoblasts. Neoblasts are heterogeneous, being comprised of multiple fate-specified classes with distinct gene expression signatures [2–4]. Most differentiated cell types have a corresponding subset of progenitors within the neoblast compartment [3,4].

Many invertebrates and vertebrates share multipotent progenitors that can give rise to neurons and glial cells (Fig 1A and [5–9]). Planarian glia have been established to be specified from "cathepsin+" or phagocytic progenitors, which are distinct from neuron progenitors [4,10–12]. This phagocytic neoblast class also produces multiple subsets of other phagocytic cells, including pigment cells. Interestingly, these progenitors have transcriptome similarity to muscle progenitors (shared *PLOD1* and *FoxF-1* expression), indicating they have a mesoderm-like identity [3,4]. Even though most glia in other organisms have been described to be of neuroepithelial origin (ectoderm), in most organisms a subset of mesodermal (e.g., microglia in vertebrates and GLR glia in *C. elegans*) or mesectoderm (midline glia in *Drosophila*) derived glial cells have also been described [13–15].

Some neoblast classes, such as for muscle, are comprised of distinct subsets, each displaying a specification program associated with a particular differentiated muscle type (e.g., longitudinal, circular, intestinal, dorsal-ventral, pharyngeal) [4,10,16]. By contrast, phagocytic specialized neoblasts do not display overt fate specification programs associated with one of their final fate outcomes [4], raising the possibility that their final fate choice is regulated at a later stage–in the migratory post-mitotic progenitors these neoblasts produce.

Planarian glia express several astrocyte-like genes, including excitatory amino acid transporters involved in the uptake of neurotransmitters, glutamine synthetase for metabolizing glutamate, and transporters for GABA and glucose [17,18]. Planarian glia are found within the neuropil of both the cephalic ganglia and the ventral cords, around commissural neurons connecting the ventral cords, in the head rim where sensory neurons are found, associated with eyes, and dorsally scattered around the animal midline (Fig 1A, [11,17,18]). Planarian neurons regenerate prior to glia formation, and various disruptions in the nervous system appear to block glia formation through unknown mechanisms [11]. Given the coordinated regeneration of planarian neurons and glia, and the lack of known glial fate specification programs in the stem cell state, we considered the possibility that some signaling mechanism between the nervous system and phagocytic progenitors could specify glia.

The Notch signaling pathway is essential for cell-fate specification and tissue patterning throughout the Metazoa. A Notch ligand, such as Delta or Jagged/Serrate, binds to a Notch receptor expressed on an adjacent cell, activating a proteolytic cascade that ultimately elicits a transcriptional response in the Notch-expressing cell [19]. This prominent signaling pathway is utilized in numerous developmental contexts, from *Drosophila* and vertebrate neurogenesis to *Drosophila* wing-disc patterning, *C. elegans* vulval development, and vertebrate somitogenesis [20–25]. In *Drosophila* neurogenesis, Notch is required to select individual cells to become neural precursors from a cluster of equipotent progenitors that express proneural genes. In the absence of Notch signaling, these progenitors retain the expression of proneural genes and

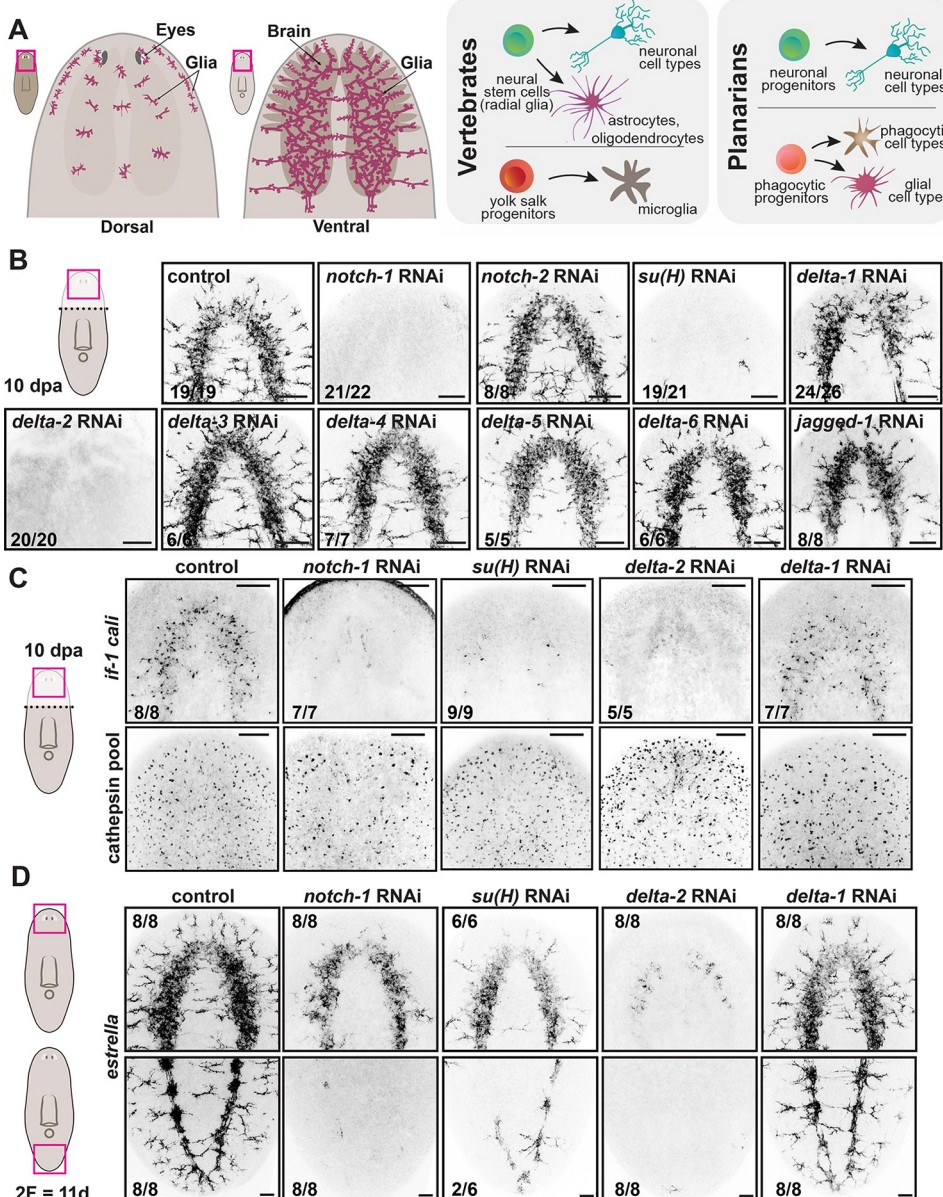

**Fig 1. Notch signaling is required for glia specification and maintenance.** (A) Cartoons show distribution of glial cells in planarians (left) and schematic lineage comparison for neurons and glia in vertebrates and planarians (right). (B) FISH shows *estrella* expression in glial cells in the neuropil following RNAi. (C) FISH shows expression of pooled markers *if-1* and *cali* (top) and pooled markers of phagocytic cells (bottom) 10 days post amputation (dpa) of different RNAi conditions. (D) FISH shows *estrella*+ glial cells in the neuropil (top) and posterior ventral cords (bottom) in uninjured short-term (2F, two feedings) RNAi animals. FISH images are representative of at least two independent experiments. Bottom left numbers indicate animals displaying the phenotype/total analyzed. Cartoons display area imaged. Anterior, up. Scale bars, 100 μm.

produce neuronal fates. Conversely, if Notch signaling is constitutively activated, progenitors do not differentiate into neurons but adopt an epidermal fate [26]. Similarly, in vertebrates, Notch signaling regulates the timing of neurogenesis and the differentiation of neural stem cells into neuronal progenitors and differentiated neurons [24,25,27–29].

Notch signaling has been implicated in regulating different aspects of gliogenesis in both *Drosophila* and vertebrates. In vertebrates, Notch can promote a gliogenic switch, ending neurogenesis [28,30]. In *Drosophila*, Notch also regulates the expression of *glial cells missing* (*gcm*), which is necessary and sufficient to specify glial cells [31,32], and can trigger glia expansion and differentiation of different glial subtypes [33–36]. Given the prominent role of Notch signaling in neuron-glia development, we sought to explore whether Notch signaling serves some role in planarian glia regeneration, distinct from its role in neuronal progenitors seen during development in many organisms. Based on our findings, we describe a model in which patterned expression of a Delta protein in particular neurons interacts with Notch-expressing, dispersed phagocytic progenitors to accomplish coordinated regeneration of planarian neurons and glia.

## Results

### Notch signaling is required for glia regeneration and maintenance

Planarians have two genes encoding Notch proteins (*notch-1* and *notch-2*), seven encoding Delta and Jagged/Serrate-family homologs (*delta-1*, *delta-2*, *delta-3*, *delta-4*, *delta-5*, *delta-6*, *jagged-1*), one gene encoding a suppressor of Hairless homolog (*su(H)*), and three *Hes*-like genes (*hesl-1*, *hesl-2*, *hesl-3*; [37]) (S1A Fig and [38]). We assessed the function of these genes using RNA interference (RNAi) and fluorescence *in situ* hybridization (FISH) to label glia using the planarian glia markers *estrella* [18], *if-1*, and *cali* [17] (Figs 1B–1D and S1B–S1E). Strikingly, inhibition of *notch-1*, *delta-2*, and *su(H)* caused a complete failure of neuropil glia regeneration following head amputation (Fig 1B and 1C). By contrast, we did not observe any obvious change in the number or pattern of other "cathepsin+"/phagocytic cells (using a pool of RNA probes specific to that cell class) after *notch-1*, *delta-2*, and *su(H)* RNAi (Fig 1C). After inhibition of these genes during regeneration, glia were also absent from the head rim, eyes, and dorsal surface, suggesting a robust role for Notch signaling in glia formation across all locations of the planarian nervous system (S1C Fig).

Glia maintenance, like regeneration, required Notch signaling. 11 days after the initiation of RNAi for *notch-1*, *delta-2*, and *su(H)*, *estrella*-expressing glia were not detectable around the eyes, head rim, dorsal midline, and ventral nerve cords of uninjured animals (Figs 1D and S1D). Some *estrella*+ cells were still present at this early time point within the neuropil of the cephalic ganglia, the region with the most abundant glia in the planarian nervous system (Fig 1D) but were mostly depleted from this region and completely absent from the head rim, eyes, and the dorsal surface at later time points (S1D Fig). Similarly, cells positive for an RNA probe pool of multiple other glia-expressing genes (*eaat2-2*, *gs*, *glut*, *if-1*, *cali*) were also diminished after inhibition of Notch signaling (S1E Fig). Some of these markers are not exclusively expressed in glial cells, explaining the remaining expression upon Notch signaling inhibition (S1F Fig). We lethally irradiated animals, which rapidly and specifically depletes neoblasts [39], and assessed the presence of glia 10 days later. *estrella*+ cells were not overtly altered in abundance by 10 days post irradiation (dpi), indicating planarian glia turnover was not rapid enough to explain their acute dependence on Notch for maintenance (S1G Fig). A caveat of this experiment is that some non-dividing glial progenitors might still be present and able to replace at least some glial cells for some time after lethal irradiation. However, because the loss of *estrella*+ cells after *delta-2* RNAi was severe (Fig 1D), a decreased number of glia after irradiation was still expected under this hypothesis. We did not detect a strong increase in TUNEL + cells within the neuropil following *delta-2* inhibition, suggesting glia were not undergoing rapid death following Notch pathway inhibition (S1H Fig). Alternatively, the acute impact of Notch pathway inhibition on glia could involve failed maintenance of glia-specific

transcription. In this scenario, some glia proteins should perdure at early RNAi time points despite mRNA transcript absence. Such a phenomenon was the case for the protein IF-1 in cephalic ganglia glia upon inhibition of Hedgehog signaling in planarians [17]. Hh controls the transcription of the planarian glial genes *if-1* and *cali* in the neuropil of cephalic ganglia [17]. Immunolabeling with an anti-IF-1 antibody showed some remaining IF-1 protein in glial cells at 11 days after initial RNAi of Notch pathway genes, indicating that at least some of these cells were still present (S1I Fig). These findings together indicate that Notch signaling is required in planarians for regeneration of glia and for maintenance of a glia gene expression program in mature glia.

## scRNA-seq shows specific loss of glia

To study the nature of the requirement of Notch signaling in the planarian nervous system and phagocytic cell regeneration and maintenance, we performed single-cell RNA sequencing (10X scRNA-seq) of regenerating (10 days post-amputation, dpa) and uninjured heads (short-term and long-term RNAi) from control, *notch-1*, and *delta-2* RNAi animals (Figs 2A and S2A). All major planarian tissue classes were present within head blastemas and uninjured heads of RNAi animals (Figs 2B and S2B). Many of the previously described subclasses of "cathepsin+"/phagocytic cells [3] were observed in heads and head blastemas of Notch-inhibited animals (Figs 2C–2F and S2C, and S1 Table). The glial cell cluster (cluster 11), by contrast, was completely absent in *notch-1* and *delta-2* RNAi blastemas (Fig 2D–2F). Glia were greatly depleted in uninjured *delta-2* RNAi animals after short-term RNAi, but only mildly reduced in the heads of *notch-1* short-term RNAi animals, similar to observations by FISH (Figs 1D, S1D, S2C and S2D). The remaining glia in short-term uninjured (st-homeostasis) *notch-1* and *delta-2* RNAi animals clustered together with control glial cells (Figs S2C and S2D). After long-term RNAi without injury (lt-homeostasis), glia were completely depleted in both *notch-1* and *delta-2* RNAi heads (Fig 2D–2F). In contrast to glia, the abundance of most "cathepsin +"/phagocytic cell subsets did not substantially and consistently change among RNAi conditions in regeneration or long-term RNAi experiments (Figs 2E, 2F and S2E). Moreover, we did not detect major changes in the percentage of neuronal clusters across the different RNAi treatments following head amputation or long-term RNAi feedings (S2F–S2H Fig and S2 Table). The absence of cells expressing known glia markers in the scRNA-seq data is taken to indicate the absence of glia in these animals; i.e., essentially no cells with detectable molecular attributes of glia regenerate or ultimately remain after long-term RNAi in uninjured animals.

## Glia loss affects histology and behavior

We next sought to investigate the impact of glia absence in the planarian nervous system. Glial cells are required for neuronal trophic support both in *Drosophila* and vertebrates, including for the uptake/recycling of neurotransmitters at synaptic clefts, for synaptic pruning, for axonal outgrowth and pathfinding, for dendrite extension, and neuronal circuit formation [13–15,40–42]. Animals were still viable and healthy after more than 30 days of *notch-1*, *su(H)*, and *delta-2* RNAi. However, we observed darkening surrounding the cephalic ganglia and ventral cords after RNAi (Fig 2G). We ruled out that darkening resulted from an overt increase in pigment cells (S2E Fig cluster 5 and S2I Fig). A dissection technique [17] that exposes cephalic ganglia and ventral cords showed the darkened region outside the neuropil, suggesting neurons might be the source of this phenomenon (Fig 2G). Darkening around neurons has previously been observed in vertebrate brain samples, but its origin is still not fully resolved [43–46]. A possible explanation suggests concentrated protein synthesis as a consequence of stress, another possibility involves dead neurons not being cleared. Further investigation is required

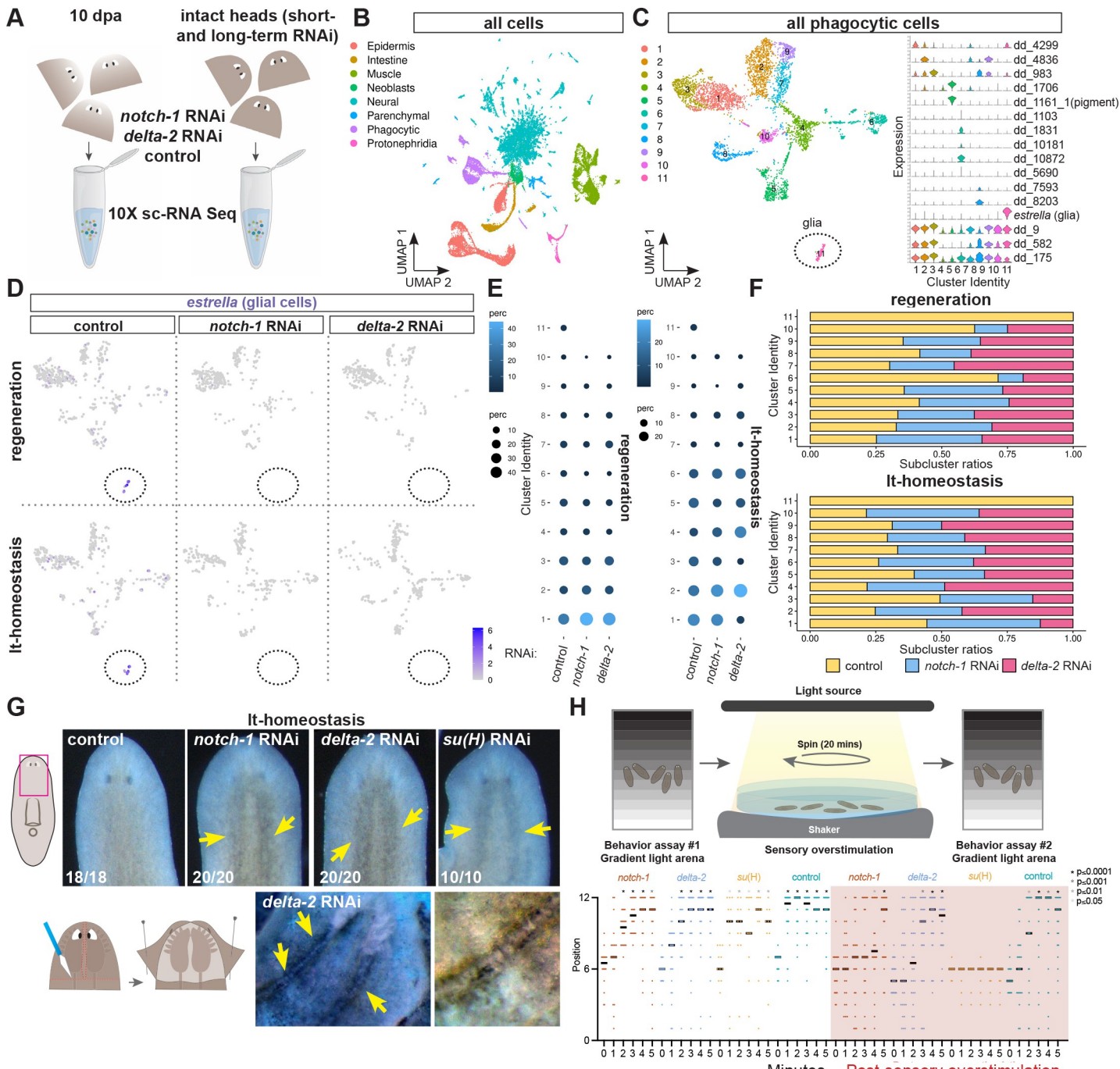

**Fig 2. scRNA-seq shows specific depletion of glia following inhibition of Notch signaling.** (A) Schematic of the 10X scRNA-seq experiments. (B) UMAP plot shows tissues represented in the dataset (all lanes). (C) UMAP plot (left) shows all phagocytic clusters in the dataset (all lanes), violin plot (right) shows expression of top or described marker for each subcluster. (D) Split UMAP plots show *estrella* expression in the different RNAi conditions and treatments. lt-homeostasis, long-term RNAi without injury. (E) Dot plots show percentage of each phagocytic subcluster in the different RNAi conditions and treatments. (F) Graph shows ratio of each phagocytic subcluster among RNAi conditions. (G) Darkening around the cephalic ganglia and ventral nerve cords in *notch-1* and *delta-2* long-term (lt) uninjured RNAi animals. Schematic of a dissection experiment to show a close up of the darkening that is excluded from the neuropil. Cartoon depicts image shown. Live images, anterior up. Images are representative of at least three experiments. Bottom left numbers indicate animals displaying the phenotype analyzed. Yellow arrows point to darkening. (H) Overstimulation (shaking and lighting) of *notch-1*, *delta-2*, and *su(H)* RNAi animals inhibits normal negative phototaxis behavior. Results from two independent experiments are shown.

to explain this phenomenon. To explore if neuron abundance and circuitry are affected by a potential stress response induced by the lack of glia, we performed immunolabeling with an anti-alpha tubulin antibody, which did not show changes in axonal projections across the RNAi treatments (S2I Fig). Moreover, percentages of all neuronal clusters and FISH for two different neuronal subsets (dd_ 21981 and dd_3524, [3]) were similarly distributed following long-term RNAi (S2H and S2I Fig). Next, to determine whether glia absence affects planarian behavior, we assessed performance in a light gradient arena [47]. All RNAi animals were able to find the darker side of the arena within 2–3 minutes, albeit slightly slower than control animals, showing the capacity of these animals for normal locomotion and negative phototaxis (S1 and S2 Movies). We next overstimulated the nervous system by continuous shaking and light stimulation to further assess the potential effects of glia loss, considering canonical functions for glia in neurotransmitter uptake and waste clearance. After stimulation, control animals displayed normal negative phototaxis but *notch-1*, *su(H)*, and *delta-2* RNAi animals were delayed or completely unable to find the darker side of the arena (Fig 2H and S3 and S4 Movies). These results suggest that the absence of glia might have an impact in negative phototaxis behavior, however, we cannot rule out a potential role for other cell types in the phenomenon observed.

### *delta-2*+ neurons guide glia formation and pattern

Although we were unable to observe *notch-1* expression by FISH, our 10X scRNA-seq dataset shows that planarian *notch-1* was strongly expressed within the phagocytic class of cells, in the glia subcluster, as well as in epidermis and protonephridia (Figs 3A, 3B, S3A and S3B). In addition, *notch-1* expression was observed in phagocytic progenitors in $G_0$ cells (post-mitotic migratory progenitors produced from neoblasts [4]) (Figs 3B and S3B). By contrast, *delta-2* expression was not observed in phagocytic cells (nor in glia), but instead, was strongly in neurons and epidermis (Figs 3A–3C and S3A–S3C). *delta-2* expression in our 10X scRNA-seq data was sparse in many neuronal clusters but clearly detected within the planarian brain by FISH (Fig 3D). These observations raised the possibility that *delta-2* acts in neurons and that *notch-1* acts in phagocytic progenitors during the regeneration of glia. This hypothesis presents an attractive candidate solution to the challenge of coordinated regeneration between neurons and glia, where contact between Delta-2 on neurons and Notch-1 on phagocytic progenitors could specify those progenitors to choose the glia fate and differentiate. Inhibition of abundant neural transcription factors (TFs), like *coe* and *sim* decreased glia numbers in the neuropil [11], and the inhibition of the TFs *ovo* and *soxB1-2*, which are required for the specification of eye cells and *pkdl-1*+ dorsal midline neurons, respectively [11,48], was previously shown to decrease numbers of glial cells in those locations without affecting regeneration of other glial subsets (S3D Fig and [11]). Indeed, we found that *delta-2* was expressed, among other neuronal subsets, in photoreceptor neurons (PRNs) in the eye, in *pkdl-1*+ cells at the dorsal midline and in neuronal clusters expressing the TFs *coe* and *sim* (Fig 3E), and that *estrella*+ glial cells were tightly associated with these *delta-2*+ neuronal classes (Fig 3E).

   *delta-2* was strongly expressed in the neuronal cluster 44 (Figs 3F and S3C) which also expressed the TFs *six1/2-2* and *zfp-3* and the marker dd_17258 (Fig 3F and S2 Table), but this cluster did not express *notch-1* (Fig 3F). dd_17258+ neurons are present in the head rim, highly concentrated at the auricles, and are presumptive sensory neurons [3]. We validated *delta-2* co-expression with dd_17258 by FISH and showed that *estrella*+ cells were tightly associated with this class of neurons (Fig 3G). We observed that some *ets-1*+; SMEDWI-1+ phagocytic progenitors existed in close proximity to *delta-2*-expressing neurons (*opsin*+ photoreceptors and dd_12758+ sensory neurons, S3E Fig). The coordinated regeneration

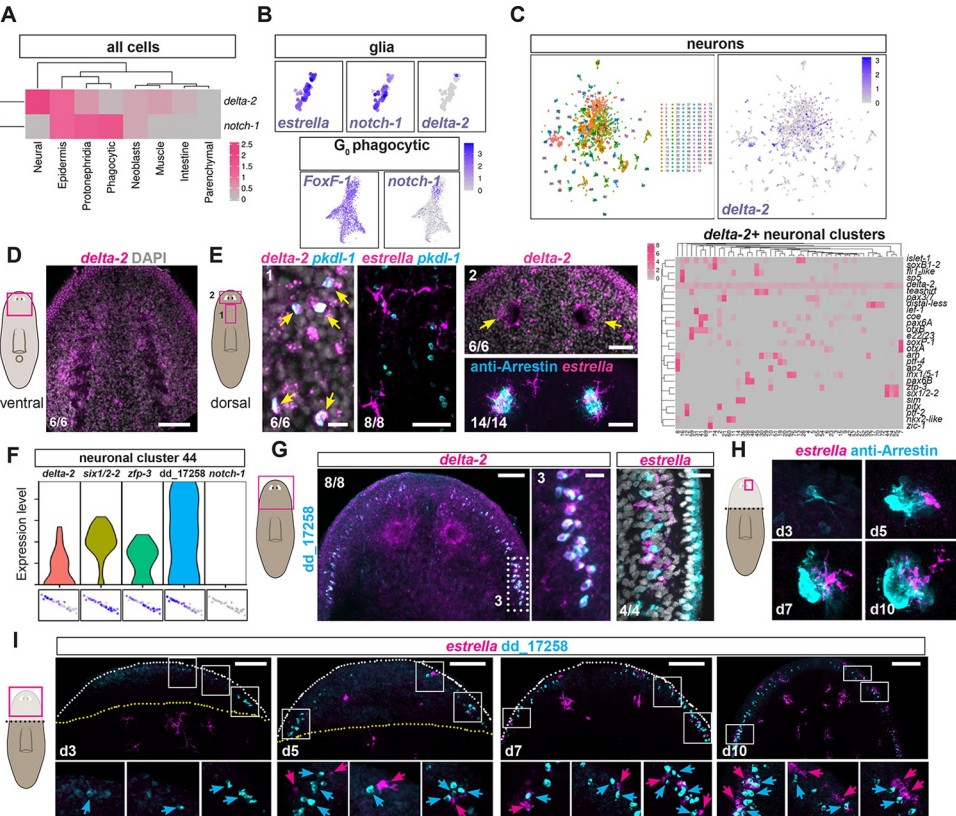

**Fig 3. *delta-2*-expressing neurons regenerate prior to *notch-1*-expressing glia.** (A) Heatmap shows expression of *notch-1* and *delta-2* in different tissues from control samples. (B) Zoom-ins of UMAP plots showing *notch-1* and *estrella* but not *delta-2* expression in the glia subcluster of control phagocytic cells (top) and *FoxF-1* and *notch-1* expression in the phagocytic progenitor subcluster (bottom). (C) UMAP plot of control neuronal clusters showing expression of *delta-2*. (D) FISH shows *delta-2* expression in the cephalic ganglia. (E) Double FISH shows co-expression (yellow arrows) of *delta-2* and *pkdl-1* in dorsal midline neurons (left) and tight association of *estrella*+ glia (magenta) and *pkdl-1* neurons (cyan, zone 1, middle). FISH shows *delta-2* expression in planarian eyes (yellow arrows, top right panel), and tight association of *estrella*+ glia (magenta) and photoreceptor neurons (Arrestin+, cyan; zone 2 right bottom). Heatmap (right) shows co-expression of *delta-2* with several neuronal TFs in different neuronal subclusters from control samples. (F) Violin plot and zoom ins of UMAP plots show expression of *delta-2* together with the TFs *six1/2-2* and *zfp-3*, and the marker dd_17258 but absence of *notch-1* in the control neuronal cluster 44. (G) FISH validates co-expression of *delta-2* and dd_17258 and tight association of *estrella*+ glial cells in zone 3. (H, I) Regeneration time course showing differentiation of neurons (Arrestin+ photoreceptors, H; and dd_17258+, I cyan) before *estrella*+ glial cells (magenta). Cyan arrows (neurons) and magenta arrows (glia) point to the close interaction between the two cell types during regeneration. Dotted white line shows animal boundary. Yellow dotted line shows plane of injury. Cartoons depict images shown. Boxes show zoom in areas. Images are representative of at least two experiments. Bottom left numbers indicate animals displaying the phenotype analyzed. Anterior, up. Scale bars, 100 μm, zoom ins 10 μm.

model also predicts that if patterned neurons are required for local differentiation of glia, then *delta-2*+ neurons should be regenerated and patterned before glia, and glia should appear where *delta-2*+ neurons form. This hypothesis is consistent with previous observations [11] and was further assessed here (Fig 3H and 3I). We analyzed the regeneration time course of Arrestin+ PRNs and sensory dd_17258+ neurons together with *estrella*+ glial cells in head blastemas following head amputation. For both of these neuronal classes, neurons were first observed within three dpa, whereas glia were first observed at five dpa (Fig 3H and 3I). Differentiated glial cells were always observed in tight association with these *delta-2*-expressing neurons, despite variability in the precise pattern of neurons, further supporting the hypothesis

that interaction between *delta-2*-expressing neurons and *notch-1*+ phagocytic progenitors is required for glia differentiation.

Inhibition of either *six1/2-2* or *zfp-3* resulted in significant decrease in the numbers of dd_17258+ cells in head blastemas (10 dpa), indicating that these TF genes were required for specification of these cells (Fig 4A), and consequently this region of the nervous system expressing Delta-2. Accordingly, there was a significant decrease in the number of *estrella*+ cells in the head rim of *six1/2-2* and *zfp-3* RNAi animals (Fig 4A), but glia in other regions of the nervous system were not compromised (Figs 4A and S4A). In cases where sparse dd_17258 + neurons were still able to regenerate under these RNAi conditions, glial cells associated with them were observed. Moreover, the TF genes *six1/2-2* and *zfp-3* were not expressed in glial cells, ruling out a direct role of these genes in the specification of glia (S4B Fig).

## Ectopic *delta-2*+ PRNs induce glia specification

To test the model that Delta-2 from neurons interacts with Notch-1 from phagocytic progenitors, we transplanted eyes from either long-term *notch-1* or *delta-2* RNAi animals–both lacking associated glia–into wild-type recipients containing normal wild-type glia progenitors (Fig 4B). We reasoned that glia should be able to differentiate in the transplant recipient upon contact with *notch-1*-deficient transplanted eyes if *notch-1* is acting in glial progenitors. By contrast, recipient wild-type glia progenitors should not be able to differentiate into glia upon contact with *delta-2*-deficient transplanted eyes, if *delta-2* is required in neurons. In both cases, no glia were present at the time of transplantation and one day thereafter (Figs 2D–2F, 4B, and S1D). Seven days post-transplantation, however, transplanted *notch-1* deficient eyes were tightly associated with glia, whereas transplanted *delta-2* deficient eyes were still depleted of glia (Fig 4B). To rule out that glia cells transdifferentiate from preexisting cells, we performed the transplantation experiments using lethally irradiated (which are depleted of neoblast cells and phagocytic progenitors) donors and recipients. In this scenario, neither *notch-1* or *delta-2* RNAi transplanted eyes were able to induce glia differentiation (S4C Fig). Taken together, these results indicate that the presence of *delta-2* in neurons is essential for *notch-1*-expressing glial progenitors to terminally differentiate into glia (model, Fig 4C). Furthermore, these findings show that placement of *delta-2*+ neurons in an ectopic location still yields access to the widely dispersed phagocytic progenitors, enabling glia formation at this location.

## Discussion

Functional nervous system regeneration requires both neurons and glia to be *de novo* produced and functionally integrated together with the correct pattern. How pattern formation for these two different cell types is coordinated in regeneration is unknown. We uncovered a mechanism by which the regenerative patterning process for neurons is leveraged for regenerating the pattern for glia. Specific neuron classes are regenerated in proper numbers and in key positions in the nervous system that activate expression of a Delta-encoding gene, *delta-2*. Delta then locally interacts with non-neural Notch-1$^+$ mesoderm-like progenitors, triggering local formation of glia from these progenitors. Ongoing Delta expression in neurons is also required to constitutively maintain the glia state in neighboring glial cells, further supporting the long-term coordination of these cell types (Fig 4C). Whereas neuron-glia choices have been found to be regulated by Notch from a common progenitor in animal development, this work shows that this same pathway can regulate glia specification and differentiation through interaction between mature neurons and an unrelated class of phagocytic progenitors. This raises the possibility that Notch has a deeply conserved role in glia specification, regardless of the progenitor source.

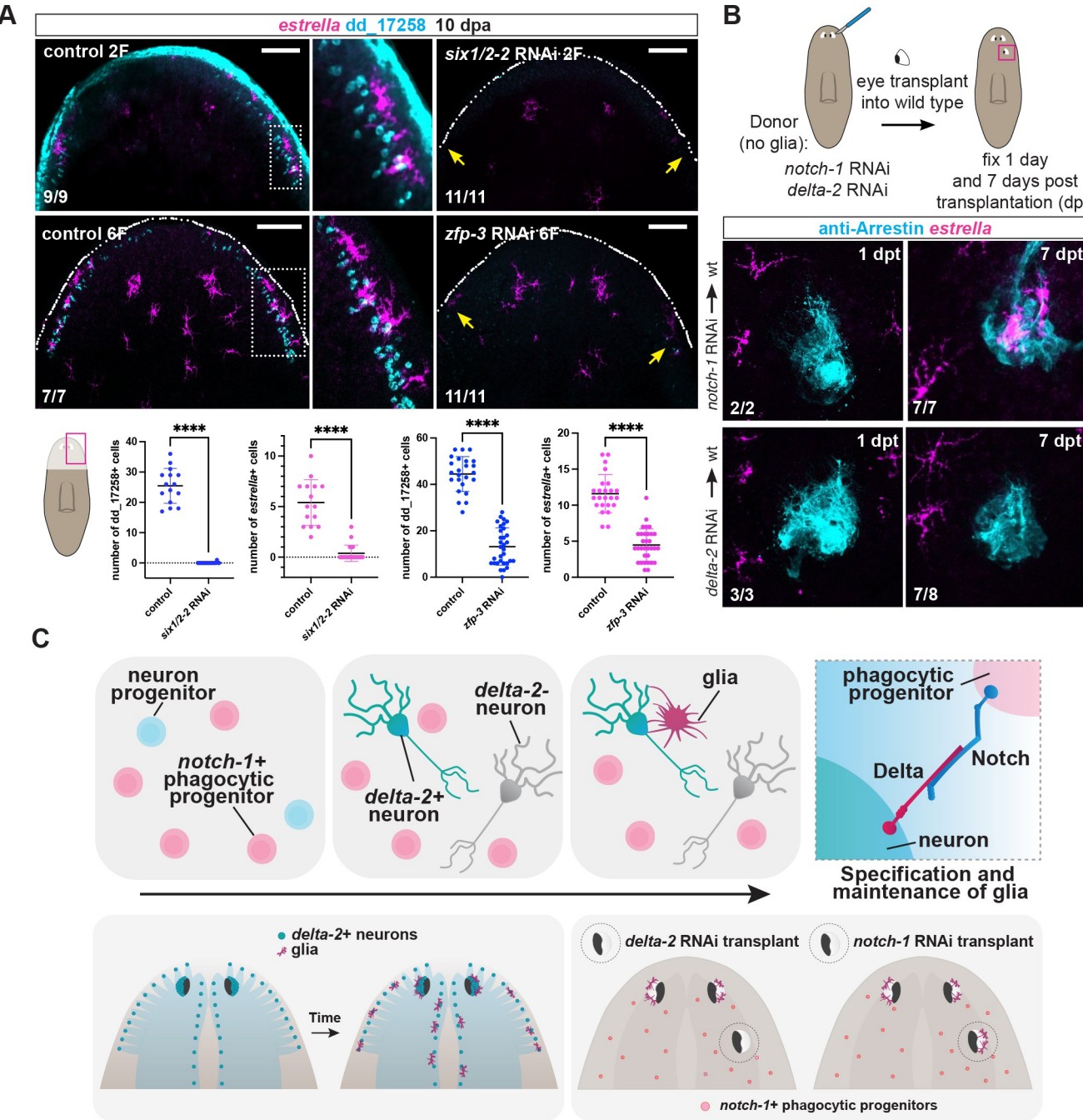

**Fig 4. *delta-2*+ neurons guide *notch-1*-mediated glia specification.** (A) Double FISH shows decreased expression of *estrella*+ glia following depletion of *delta-2*-expressing dd_17258+ neurons after *six1/2-2* and *zfp-3* RNAi. Yellow arrows point to missing neurons and glia. Dotted white line shows animal boundary. Graphs (bottom) show mean ± SD, analyzed by unpaired Student's t-test, p-value< 0.0001. (B) Top: Schematics of the experiment. Bottom: FISH and immunostainings show specification of *estrella*+ glial cells (magenta) associated with Arrestin+ photoreceptor neurons (cyan) at seven days post-transplantation (dpt) of *notch-1* RNAi eyes but not *delta-2* RNAi eyes. No glia were observed in any condition one day post-transplantation. (C) Model: Coordinated regeneration of glia and neurons involves *delta-2*+ neurons guiding differentiation of *notch-1*+ phagocytic progenitors into glia. Differentiation of *delta-2*+ neurons occurs prior to glia differentiation. Ectopic *delta-2*+ neurons are sufficient to induce ectopic glia differentiation. Cartoon depicts area analyzed. Boxes show zoom in areas. Images in (A) are representative of at least two experiments, and in (B) of one experiment. Bottom left numbers indicate animals displaying the phenotype analyzed. Anterior, up. Scale bars, 100 μm.

How the precise numbers and positions for interacting cells produced from different progenitors can be matched is an important problem in regeneration. In the case of neurons and glia, the exact number and position of certain neuron classes can be variable animal-to-animal, or even between the left-to-right sides of the animals (e.g., see Fig 3G and 3I). Furthermore, glia are produced by a progenitor class that must differentiate at a variety of distinct positions. By having the final fate choice for glia be determined in migratory post-mitotic phagocytic progenitors, local signals from neurons interacting with these progenitors can solve these challenges, resulting in precise local coordination of pattern. The modular nature of this coordinated regeneration mechanism could also readily facilitate evolution of new configurations of glia in the nervous system, by any particular neuron population acquiring or losing expression of *delta-2* and having access to dispersed phagocytic progenitors. This mechanism uncovers a novel approach for integrating glia with neurons in a regenerating nervous system, solving the challenge of specifying glia in proper numbers and locations utilizing a Delta-Notch glia-specification signaling process.

## Materials and Methods

### Ethics statement

No vertebrate animals were used in this research.

### Animal husbandry

*Schmidtea mediterranea* clonal asexual strain CIW4 animals, starved for 7–14 days before experimentation, were used for all experiments. All animals utilized were healthy, not previously used in other procedures, and were of wild-type genotype. Animals were cultured in plastic containers or petri dishes for experiments, in 1x Montjuic water (1.6 mmol/l NaCl, 1.0 mmol/l $CaCl_2$, 1.0 mmol/l $MgSO_4$, 0.1 mmol/l $MgCl_2$, 0.1 mmol/l KCl and 1.2 mmol/l $NaHCO_3$ prepared in Milli-Q water) at 20 ˚C in the dark.

### Replication, size estimation, and randomization

The number of independent experiments performed is indicated in Figure legends; numbers of animals used in each experiment are indicated in each panel. No sample size estimation was performed. Animals for all experiments were randomly selected from a large collection of clonal animals. All animals were included in statistical analyses, without exclusions. Images were not randomized.

### 10X single-cell RNA sequencing and analysis

Cell samples were collected from 8–10 animals and pooled together for homeostasis conditions, and 10–12 head blastemas for regeneration. In all cases, the headpiece anterior to the auricles was collected. RNAi animals were fed six times, heads amputated, and head blastemas harvested after ten days. For short-term homeostasis experiments, animals were fed twice over a week, and heads were harvested ten days after the first RNAi feeding. For long-term homeostasis experiments, animals were fed six times over three weeks, and two more times once a week. Heads were harvested six weeks after the first RNAi feeding. Tissue fragments were gently amputated for dissociation using a scalpel and collected in planarian water. The fragments were then briefly washed with DPBS without calcium or magnesium (Gibco, Thermo-Fisher Scientific, Cat. No: 14190250), placed in Papain solution, and prepared with Neurobasal Medium (Gibco, Thermo-Fisher Scientific, Cat. Num.: 21103049) supplemented with B-27 Plus Supplement (Gibco, Thermo-Fisher Scientific, Cat. Num: A3582801) and incubated in a

34C water bath for 20 mins. Following gentle pipetting, to dissociate the fragments into single cells, a second 6-minute-long incubation, and additional pipetting were performed. Papain was inhibited using an Ovomucoid Inhibition Solution following the manufacturer's recommendations. The samples were then centrifuged at 500 g for 5 min, resuspended in ice-cold CMF solution calcium-magnesium-free solution with 1% BSA (CMFB) and filtered using a 40μm filter (400 mg/L $NaH_2PO_4$, 800 mg/L NaCl, 1200 mg/L KCl, 800 mg/L $NaHCO_3$, 240 mg/L glucose, 1% BSA, 15 mM HEPES, pH7.3). Samples were resuspended in ice-cold CMFB and counted with Trypan blue to determine the optimal number of cells needed for the 10X single-cell sequencing procedure. Each condition was run as a single sample in 10X library preparation. Cells were processed by the WIGTC core (Whitehead Institute for Biomedical Research) using 10X Genomics Chromium Controller and Chromium single cell 3′ Library & Gel Bead Kit (PN 1000006) following standard manufacturer's protocol. Samples were sequenced on an Illumina NovaSeq 6000 (150 × 150 paired-end reads) across all lines. Sequencing reads were mapped using a GTF file of Smed_v6 (https://planmine.mpinat.mpg. de/planmine/model/bulkdata/dd_Smed_v6.pcf.contigs.fasta.zip) genes in the context of the Smes_g4 (https://planmine.mpinat.mpg.de/planmine/model/bulkdata/dd_Smes_g4.fasta.zip) genome. This GTF file was generated by using BLAT to map all Smed_v6 transcripts to the Smes_g4 genome and each transcript was assigned to a single genome location based on the best alignment score. Transcripts were then collapsed using genome location before mapping using the Cell Ranger 7.2.0 pipeline. Cells were assessed for nUMI, nGene, and percent mitochondrial transcript content, which was represented in violin plots (S2A Fig). Percent mitochondrial content was based on mitochondrial genes reported in [49] which are represented in v_6 of the Dresden transcriptome (*dd_Smed_v6_258_0_1*, *dd_Smed_v6_289_0_1*, *dd_Smed_v6_292_0_1*, *dd_Smed_v6_297_0_1*, *dd_Smed_v6_344_0_1*, *dd_Smed_v6_505_0_1*, *dd_Smed_v6_753_0_1*, *dd_Smed_v6_957_0_1*) and on the highly abundant mitochondrial transcripts (mtRNA_1, mtRNA_2) from [50]. Doublets were identified via scDblFinder (https://bioconductor.org/packages/release/bioc/html/scDblFinder.html) and removed after basic QC filtering; any cells with nFeature_RNA < 750, nFeature_RNA > 3000, nCount_RNA < 1000, or nCount_RNA > 20000 were removed from the dataset prior to analysis. 10X analysis was performed using Seurat 5.1.0 [51] where cells were visualized using the uniform manifold approximation and projection (UMAP) algorithm. The number of dimensions used with RunPCA, RunUMAP, and FindNeighbors was determined using JackStraw with a p value cutoff of 0.05. Clusters were determined via FindClusters using the leiden algorithm. UMAP plots of identities (all cells in Figs 2B / S2B, all phagocytic cells in Figs 2C / S2C, control neuron cells in Fig 3C, all neuron cells in S2F Fig, control cells in S3B Fig, $G_0$ migratory progenitors in S3B Fig, and control phagocytic cells in S3B Fig) were created using Seurat's DimPlot function. UMAP plots of gene expression (phagocytic cells in Fig 2D, control glial cells / $G_0$ phagocytic progenitors in Fig 3B, control neurons in Fig 3C, control neurons cluster 44 in Fig 3F, all cells in S2B Fig, control phagocytic cells in S1F Fig, all control cells, $G_0$ phagocytic progenitors and control phagocytic cells in S3B Fig, control glia cells in S4B Fig) were created using Seurat's FeaturePlot function. Marker genes were identified using control (*unc-22*) RNAi cells with Seurat's FindMarkers function (phagocytic markers–S1 Table, neural markers–S2 Table). For a heatmap of top markers from control cells in neural clusters (S2G Fig), markers were filtered with pct.1 > 0.4 and pct.2 < 0.4, sorted in order of smallest p_val_adj, and the top marker for each cluster was taken. For a heatmap (control phagocytic cells in S3A Fig) and violin plot (Fig 2C) of all cells in phagocytic clusters, markers were cross-referenced with existing literature to select known marker genes. The gene *aqp-1* was not identified via FindMarkers but was included in the heatmap and violin plot. For a heatmap of *delta-2*+ neuronal clusters (Fig 3E), control neuron cells were subset using Seurat's WhichCells to those

with the expression of *delta-2* > 1.2 and markers were selected using known transcription factors from existing literature. Heatmaps of all control cells were created using select genes rather than those from FindMarkers (Figs 3A and S3A). All heatmaps were constructed using genes found as previously described, obtaining a group's average expression by Seurat's AverageExpression function, scaling data by genes using R's base scale() function with center = F, and plotting via ComplexHeatmap::Heatmap [52]. Violin plots of control neuron cells from cluster 44 (Fig 3F), all control neuron cells (S3C Fig), and QC stats for all lanes (S2A Fig) were created using Seurat's VlnPlot function. A custom violin plot function was made for the violin plot of all phagocytic clusters (Fig 2C). Bar plots of ratios of each of the phagocytic subclusters or of cell types among the RNAi treatments were created by first sampling cells from each RNAi condition, using a number of cells equal to the least populated condition. These ratios were then plotted using ggpubr::ggbarplot https://rpkgs.datanovia.com/ggpubr/ (Figs 2F, S2B and S2D). Dotplots of ratios of seurat clusters within a given RNAi treatment were created using ggplot2 with geom_point (Figs 2E, S2D and S2H).

### Protein domain analysis

The longest predicted ORF of the Notch signaling components from the dd_v6 transcriptome assembly was inspected for domain architecture similarities using SMART with HMMER searches of Outlier homologs, PFAM domains, and signal peptide prediction. The protein domain structure of the genes utilized for RNAi is graphically represented (S1A Fig).

### Gene cloning

All constructs were cloned from cDNA into the pGEM vector (Promega). These constructs were used to synthesize RNA probes and dsRNA for RNAi experiments. All genes used are identified with a Smed_v6_dd contig id that can be found online at https://planmine.mpinat.mpg.de/planmine/begin.do. The following primers were used: *notch-1* (dd_12732) fw: 5'-ccctttcaatgccagcctt-3' rv: 5'-gtcggcaatgtgaaaaacct-3'; *notch-2* (dd_7067) fw: 5'-gtcggcaatgt-gaaaaacct-3' rv: 5'- tcgcgtctcacaatacttgc- 3'; *delta-1* (dd_10221; previously named *jagged-like* [18]) fw: 5'-caagagctgggtttcgatgc- 3' rv: 5'- tccgaagcatccgatcatga- 3'; *delta-2* (dd_15985; previously named *delta-like [48]* fw: 5'-ggactggcagcaattgtgaa-3' rv: 5'-cggcacagctgtagaaattct-3'; *delta-3* (dd_13823) fw: 5'-tctctgctgtggaaattgca-3' rv: 5'-acaacaccgagataagggca-3'; *delta-4* (dd_6929) fw: 5'-tcaggaaagacagataatggtgt-3' rv: 5'-agtgccttgacgattccatt-3';; *delta-5* (dd_9663) fw: 5'-gactacaacgcccaatatgca- 3' rv: 5'-tgacctatcttgcaaacggt-3'; *delta-6* (dd_4792; previously named jagged-2-1 [53]) fw: 5'- agtctccggcgaaagtgaat- 3' rv: 5'- catgtagccagaccgaggat- 3'; *jagged-1* (dd_11876) fw: 5'-ctggcacaaactgcgagata-3' rv: 5'-tcggattgaatgggaaagag-3 '; *Su(H)* (dd_10150/dd_6047) fw: 5'-acacaactgctgcaaactgt-3' rv: 5'-tttacaggcacctctaccgg-3'; *hesl-1* (dd_32934) fw: 5'-aaatggaaagacgacgaagg-3' rv: 5'-catcattcataaggcttccaaa-3'; *hesl-2* (dd_17332) fw: 5'-ccattccaaatc-gaatctcaa-3' rv: 5'-gacatttcccaaggtctcca-3'; *hesl-3* (dd_22479) fw: 5'-agcaacaatccagaactcga-3' rv: 5'-aagggaggagatggaggaga-3'. Previously cloned genes were used: *zfp-3* (dd_36433, [54]); *soxB1-2* (dd_8104, [11]); *ovo* (dd_48430, [48]); *six-1/2-2* (dd_9774, [55]); dd_17258 [3], *pkdl-1* (dd_13975, [3]); *estrella* (dd_1792, [18]); intermediate filament-1 (dd_12254 [17]); *calamari* (dd_9961 [17]) for the glia pool: (*if-1*; *cali*; *gs* dd_896; *glut* dd_5447; *eaat2-2* dd_1106, *trpm* dd_3620, [17, 18]) and for the cathepsin pool (dd_7593, dd_6149, dd_1260, dd_10872, dd_1103, [3])

### RNAi

For RNAi experiments, dsRNA was synthesized by *in vitro* transcription reactions (Promega) using PCR-generated templates with flanking T7 promoters, followed by ethanol precipitation

and annealed after resuspension in water. The concentration of dsRNA varied in each prep between 4 and 7 µg/ml. dsRNA was then mixed with planarian food (liver) and 2 µl of this mixture per animal (liver containing dsRNA) was used for feedings. Animals in regeneration experiments were fed six times in three weeks. Animals were then amputated the day after the last feeding. Animals in short-term homeostasis experiments were fed twice a week. Animals in long-term homeostasis experiments were fed 8–10 times during 4–6 weeks.

### Fluorescence in situ hybridizations, immunostainings, and TUNEL assay

Animals were killed in 5% NAC, fixed in 4% formaldehyde solution in PBST (PBS with 1% Triton-X) as described [49]. For FISH, animals were rehydrated, bleached, and treated with proteinase K (2 µg/ml) as described [49]. Following overnight hybridizations, samples were washed twice in pre-hybridization buffer, 1:1 pre-hybridization-2x SSC, 2x SSC, 0.2x SSC, and PBST. Subsequently, blocking was performed in 10% Western Blocking Reagent (Roche, 11921673001) PBST solution for DIG probes, or 5% Horse serum and 5% Western Blocking Reagent for FITC probes. Antibody washes were then performed for one hour followed by tyramide development. Peroxidase inactivation with 1% sodium azide was done for 90 min at room temperature. Alpha-tubulin antibody (1:500, MS581) labeling was then performed as described [56]. Brightfield images were taken with a Zeiss Discovery Microscope. Fluorescent images were taken with a Leica Stellaris confocal microscope. Co-localization analyses of FISH signals were performed using Fiji/ImageJ. For each channel, histograms of fluorescence intensity were used to determine the cut-off between signal and background. All FISH images shown are representative of all images taken in each condition and are maximal intensity projections, except otherwise indicated. All images, unless otherwise indicated, are anterior up. For TUNEL labeling, fixed animals were bleached overnight at room temperature in $H_2O_2$ (Sigma, 6% in PBSTx), incubated for 10 min in Proteinase K solution (2 µg/ml in PBSTx with 0.1% SDS), and post-fixed in formaldehyde (4% in PBSTx). Single animals were transferred to a 96-well U-bottom plate. PBSTx was replaced with a 20 µL reaction mix (3 parts ApopTag TdT enzyme mix, 7 parts ApopTag reaction buffer), and incubated overnight at 37 ˚C. Animals were then washed in PBSTx followed by development in a 20 µL development solution (one part blocking solution, one part ApopTag anti-digoxigenin rhodamine conjugate), and incubated in the dark at room temperature overnight. Samples were washed in PBSTx and counterstained with DAPI (Sigma, 1 µg/ml in PBSTx). TUNEL was performed using reagents from the ApopTag Red in Situ Apoptosis Detection Kit (Millipore, #S7165).

### Irradiation

Animals were irradiated using a dual Gammacell-40 cesium source set to deliver 6,000 rads. Animals were fixed 10 days after irradiation.

### Surgical procedures

For all surgical procedures, animals were placed on moist filter paper on a cold block to limit movement. Eye transplants were performed as described [47]. Briefly, eyes from *notch-1* and *delta-2* RNAi animals were surgically removed and transplanted into wild-type animals. Transplanted animals were then immobilized using Type IV, 5% ultra-low melting agarose (Sigma), and the solidified gel was covered with filter Whatman paper (GE Healthcare, Life Sciences) soaked in Holtfreter's Solution. Animals were left at 10 C overnight and rescued the following day by cutting the surrounding gel and transferring them into planarian water. To test transdifferentiation, *notch-1* or *delta-2* RNAi animals were lethally irradiated four days after the last RNAi feeding, and the eyes were transplanted at day 5 post-irradiation into day 5

post-irradiation wildtype recipients. Recipients were then fixed at day 5 post-transplantation and analyzed for presence of glial cells.

## Behavioral assays

Behavior assays were performed as described [47]. Briefly, a gradient layout for the behavior arena was generated using Adobe Illustrator CC software. The gradient arena was displayed on a horizontally placed iPad continuously. A rectangular one-well plate containing planarian water was placed on top of the iPad and the arena was covered with a box to eliminate any directional light from the test environment. An iPhone was placed on top of the box to record videos of the behaving animals. Animals were placed in positions 5 and 6 of the arena at the start of each trial. Positions of each animal at the end of each minute were recorded for a total of 5 minutes. Time 0 values were not analyzed for significant deviation from random distribution. For over-stimulation, animals were placed on an orbital shaker (VWR) at a low setting (between 50 and 100 rpm) and were shaken for 20 minutes under a direct light from a LED light source positioned 25–30 cm away.

## Quantifications and statistical analysis

Cell numbers from FISH were manually counted and comparisons between the two groups were performed using unpaired Student's $t$-test analyses (Prism software). Graphs show the mean and standard deviation. For behavior analysis, the positions of each animal at the end of each minute were averaged and compared for each time point for all groups. One-sample t-test was performed comparing each time point's mean with a hypothetical value of 6, corresponding to chance. Bonferroni's correction was applied. In order to test if the number of experimental RNAi cells observed within a given phagocytic cluster was significantly different than expected, we performed a hypergeometric test for over representation or under representation using R's stats::phyper function and implemented a significance threshold of $p < 0.0002$. Binomial confidence intervals (95% CI) we are also calculated for expected and observed ratios of perturbed cells to total cells using R's stats::binom.test function where p, the hypothesized probability of success, is equal to the expected ratio (total experimental RNAi cells / total cells). Results were plotted using ggplot2 with geom_point and geom_errorbar.

## Supporting information

**S1 Fig. Notch signaling is required for glia regeneration and maintenance.** (**A**) Protein domain structure of Notch signaling components used in this study. (**B-E**) FISH shows glial cell expression during regeneration (**B,C**) and in uninjured animals (**D, E**) following different RNAi conditions. (**F**) UMAP plots show expression of glia markers used in S1E Fig and *estrella* within the phagocytic cell clusters. (**G**) *estrella*+ glia expression in irradiated uninjured animals (dorsal view top, ventral view bottom). (**H**) TUNEL staining shows a similar number of apoptotic cells within the neuropil. (**I**) Some IF protein is still present after RNAi treatment. FISH images are representative of one experiment (E,G,I) and at least two independent experiments (B-D, H). Numbers indicate animals displaying the phenotype shown. Cartoons display area imaged, dark animal cartoon indicates dorsal view, light shade cartoon indicates ventral view. Anterior, up. Scale bars, 100 μm.
(PDF)

**S2 Fig. scRNA-seq shows specific loss of glia following inhibition of the Notch signaling pathway.** (**A**) QC for all lanes. Violin plots show number of UMIs (left) and genes (center) per cell in each of the sequencing lanes. st, short-term; lt, long-term RNAi treatments.

Table (right) shows total number of cells after QC per lane, and mean reads per cell. (**B**) UMAP plot (top left) shows all cell clusters. UMAP plots show expression of specific tissue markers. UMAP plots show contribution of cells to different clusters per treatment (bottom left) and RNAi condition (bottom center). Graph (bottom right) shows contribution of every lane to each tissue. (**C**) Split UMAP plots show contribution of cells to different phagocytic clusters per treatment (top) and per RNAi condition (bottom). (**D**) Split UMAP plots (left) show *estrella* expression, dot plot (center) shows percentage of each phagocytic cluster, and bar graph (right) shows proportion of each phagocytic cluster in short-term homeostasis per RNAi condition. (**E**) Dot plot of ratios of perturbed cells to total cells for phagocytic clusters. Black symbols indicate expected ratios from all cells. Colored symbols indicate observed ratios from cells of a given cluster (blue = not significant, red = significant, $p < 0.0002$). (**F**) UMAP plot shows neuronal clusters for all lanes. (**G**) Heatmap shows top specific markers for each neuronal cluster. (**H**) Dot plots show percentage of each neuronal cluster in regeneration and lt-homeostasis in each RNAi condition. (**I**) FISH (top) shows no overt difference in pigment cells among different RNAi conditions after long-term RNAi. Immunostaining (bottom) shows normal axonal projections and neuronal numbers in long-term homeostasis RNAi animals. FISH images are representative of one experiment. Numbers indicate animals displaying the phenotype shown. Cartoon displays area imaged. Anterior, up. Light shade cartoon indicates ventral view. Scale bars, 100 μm.
(PDF)

**S3 Fig. *notch-1* is expressed in glia and phagocytic progenitors whereas *delta-2* is strongly expressed in neurons.** (**A**) Heatmaps show expression of Notch signaling components in different tissues (top) and in phagocytic clusters (bottom). (**B**) UMAP plots show expression of (top) *notch-1* and *delta-2* in all cells from control RNAi conditions, (center) *FoxF-1* and *notch-1* in G0 migratory progenitors [4], and (bottom) *notch-1* and *estrella* in phagocytic clusters. Left UMAP plots in each row show contribution of cells to different respective clusters. (**C**) Violin plot shows expression of *delta-2* in neuronal clusters from control RNAi conditions. (**D**) Immunostaining and FISH (top row) show expression of *estrella*+ glial cells (magenta) and photoreceptor neurons (Arrestin+, cyan) in different RNAi conditions. Double FISH (bottom row) show expression of *estrella*+ glial cells (magenta) and *pkdl-1*+ neurons (cyan) in different RNAi conditions. Left and center panels, dorsal view; right panels, ventral view. Yellow arrows point to eye glia (top) and dorsal midline glia (bottom). FISH images are representative of two independent experiments. Numbers indicate animals displaying the phenotype shown. Graphs on the right show mean ± SD, analyzed by unpaired Student's t-test, p-value< 0.0001. (**E**) FISH and immunostaining shows proximity of *ets-1*+SMEDWI-1+ cells to *delta-2*-expressing neurons (*opsin*+ photoreceptors and dd_17258+ sensory neurons). Cartoons displays area imaged. Anterior, up. Scale bars, 100 μm (D), 10 μm (E).
(PDF)

**S4 Fig. Inhibition of *six1/2-2* and *zfp-3* does not affect glia formation in the eye, dorsal midline and neuropil.** (**A**) FISH shows *estrella*+ glial cells in different locations. Top row, dorsal view. Center and bottom rows, ventral view. FISH images are representative of two independent experiments. Numbers indicate animals displaying the phenotype shown. 2F, two RNAi feedings; 6F, six RNAi feedings. (**B**) UMAP plots and FISH show lack of expression of the TF *six1/2-2* and *zfp-3* in the glia cluster (*estrella*+ cells). (**C**) Cartoons on the left summarize the experimental setup. FISH shows that glia can differentiate after day 5 post eye transplantation of *notch-1* RNAi animals into wildtype recipients. However, no glia differentiation is observed if eye transplant donors and wildtype recipients were lethally irradiated. Cartoons display area imaged, dark animal cartoon indicates dorsal view, light shade cartoon indicates

ventral view. Anterior, up. Scale bars, 100 μm (A), 10 μm (B), 50 μm (C).
(PDF)

**S1 Table. Markers for each phagocytic cell cluster.**
(XLS)

**S2 Table. Markers for each neuronal cell cluster.**
(XLS)

**S1 Movie. Behavior of control, *notch-1* and *delta-2* RNAi animals pre-stimulation.**
(MP4)

**S2 Movie. Behavior of *su(H)* RNAi animals pre-stimulation.**
(MP4)

**S3 Movie. Behavior of control, *notch-1* and *delta-2* RNAi animals post-stimulation.**
(MP4)

**S4 Movie. Behavior of *su(H)* RNAi animals post-stimulation.**
(MP4)

## Acknowledgments

The authors thank members of the Reddien lab for discussions and comments on the manuscript.

## Author Contributions

**Conceptualization:** M. Lucila Scimone, Peter W. Reddien.

**Formal analysis:** M. Lucila Scimone, Patrick Aoude, Kutay D. Atabay.

**Funding acquisition:** Peter W. Reddien.

**Investigation:** M. Lucila Scimone, Bryanna Isela-Inez Canales, Patrick Aoude, Kutay D. Atabay.

**Methodology:** M. Lucila Scimone, Bryanna Isela-Inez Canales, Patrick Aoude, Kutay D. Atabay, Peter W. Reddien.

**Project administration:** M. Lucila Scimone, Peter W. Reddien.

**Software:** Patrick Aoude.

**Supervision:** M. Lucila Scimone, Peter W. Reddien.

**Validation:** M. Lucila Scimone, Bryanna Isela-Inez Canales, Patrick Aoude, Kutay D. Atabay.

**Visualization:** M. Lucila Scimone, Kutay D. Atabay, Peter W. Reddien.

**Writing – original draft:** M. Lucila Scimone, Bryanna Isela-Inez Canales, Patrick Aoude, Kutay D. Atabay, Peter W. Reddien.

**Writing – review & editing:** M. Lucila Scimone, Bryanna Isela-Inez Canales, Patrick Aoude, Kutay D. Atabay, Peter W. Reddien.

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
