## [Decision Letter · Decision Letter 0]

17 Sep 2024

Dear Dr Reddien,

Thank you very much for submitting your Research Article entitled 'Coordinated neuron-glia regeneration through Notch signaling in planarians' to PLOS Genetics.

The manuscript was fully evaluated at the editorial level and by independent peer reviewers. The reviewers appreciated the attention to an important problem, but raised some substantial concerns about the current manuscript. Based on the reviews, we will not be able to accept this version of the manuscript, but we encourage you to submit a much-revised version. We cannot, of course, promise publication at that time.

Reviewer 3 raised concerns that would need additional experiments, some of which you likely already have done, and would focus your attention on reviewer 3 comments below. Minor concerns regarding mislabeling of figures, and fonts being too small to read should also be addressed. 

Additionally, the scRNAseq data is set to set to not be released until Sept. 2026. Should your manuscript be accepted for publication in PLOS Genetics, we ask that these data be made available to the public at the time of publication. Please add a data availability statement to this effect when submitting revisions. 

If you decide to revise the manuscript for further consideration at PLOS Genetics, please aim to resubmit within the next 60 days, unless it will take extra time to address the concerns of the reviewers, in which case we would appreciate an expected resubmission date by email to plosgenetics@plos.org.

To resubmit, log into your Editorial Manager account and select the option 'Revise Submission' in the 'Submissions Needing Revision' folder.

We are sorry that we cannot be more positive about your manuscript at this stage. Please do not hesitate to contact us if you have any concerns or questions.

Yours sincerely,

Giovanni Bosco, Ph.D.

Section Editor

PLOS Genetics

Giovanni Bosco

Section Editor

PLOS Genetics

Reviewer's Responses to Questions

**Comments to the Authors:**

Reviewer #1: Functional regeneration requires the restoration of different cell types in the correct place and abundance. In highly-regenerative planarians, neurons and glia are regenerated from distinct progenitors. This study investigates the role of Delta and Notch signaling to specify glia. The authors demonstrate that RNAi knockdown of notch-1 or delta-2 leads to an absence of glia, both after regeneration and in uninjured planaria. They employ single cell RNA sequencing to hypothesize that delta-2 expressed by neurons and notch-1 expressed by glial progenitors is required for glial differentiation. They support this hypothesis by depleting specific neuronal populations and observing a loss of only the associated glia. In addition, through transplantation experiments they demonstrate that delta-2-expressing photoreceptor neurons are sufficient for ectopic glial formation. Taken together, this is a convincing study with broad relevance to the field of regeneration that supports the model that Delta-2+neurons drive glial progenitor differentiation and glial maintenance via Notch-1.

Minor comments

1) While most biologists will be familiar with notch signaling, including a sentence that summarizes the pathway and how Notch and Delta proteins function would provide important context.

2) The authors should double check their references in the text to figure 2 as they are mislabeled in the text.

3) It is difficult to see the TUNEL+ cells in figure S1G. Could the authors show this channel independently of the DAPI?

4) A lot of the labels on figures related to scRNAseq are too small to be readable. In addition, figure 3C is too small to be interpretable.

5) The authors knockdown notch-1, delta-2, or su(H) and observe deficits in negative phototaxis behavior. In the text, they write “to determine whether glia absence affects behavior”, yet it is possible with these manipulations there are deficits independent of glial absence that could be impacting behavior too.

5) The authors should include labels in Fig 3E for the colors of the image.

5) The authors state that the TF-encoding genes six1/2-2 and zfp-3 were not expressed in glial cells, but they cite their scRNAseq data as evidence of this lack of expression. While this approach can argue for expression of a gene in a specific cell-type, the opposite may not be true due to sequencing depth. To make this claim, the authors should do in situs or otherwise validate in an independent way that these transcription factors are not in glia.

Reviewer #2: This is a truly remarkable and elegant study showing that planarian neurons induce terminal differentiation of glial progenitors via Delta/Notch signaling. Planarian neurons are known to form prior to glia in regeneration and both cell types differentiate from distinct progenitor cell types, so how the animal can precisely coordinate the production of these closely associated cell types remained unclear. First, the authors show that delta-2 or notch-1 RNAi caused failure of glial differentiation and maintenance of gene expression, identifying new regulators of this cell type. Expression of these factors was selective to several types of neurons (delta-2+) and glia (notch-1+), suggesting a potentially direct interaction mediating the decision to terminally differentiate glia. This included examples of photoreceptor neurons along with associated glial cell types from the planarian eye, which is a tractable and simple organ system. Eyes from either notch-1 or delta-2 RNAi animals lacked associated glia, while transplantation of eyes from notch-1(RNAi) animals into wild-type animals, but not from delta-2(RNAi) animals into wild-type animals, enabled the formation of glia onto the transplanted photoreceptor neurons. Therefore, the authors' experiments nicely argue that delta-2+ neurons instructively, and likely directly, commit the differentiation of eye-associated notch-1+ glial cells. Along the way, the authors develop an entirely new sensitized behavioral assay to show that glia are essential for planarians to robustly avoid light after experiencing a sensory overstimulation. They also find, intriguingly, that the planarian nervous system exhibits a darkening of tissue when glia are lost, a mysterious phenomenon not yet fully explained but present across species, suggesting planarians could also be used to understand the function and pathology of this condition. Together, their findings reveal a mechanism that enables the coordinated differentiation of associated cell types from distinct progenitors so that final organs attain appropriate numbers of each cell type.

This paper was a joy to read, an “instant classic.” The experiments are beautifully executed, rigorously supportive, and introduce several new methods of analysis in pursuit of the question at hand. The message is broad and important. I don’t believe there are any significant experiments further needed to support the conclusions of this study.

Minor comments:

Suggest adding information to the cartoon in S1F indicating irradiation and dorsal versus ventral so that the figure is more clearly understood without the legend.

Authors’ discretion- For the Fig 2G data, I personally find it easier to read data left to right when controls are indicated first on the left followed by experimentals on the right for each set of treatments.

Fig 2G it wasn’t clear to me how the statistical test was done or which test was done. I assume authors used t-tests between t=0 and each timepoint within each condition?

There is a figure callout issue at Figure 2G bottom of page 11 (also please add page numbers). “However, we observed darkening…” should cite to Fig 2F not Fig 2G. Likewise callouts to Fig 2H should be 2G. Earlier callouts in Fig 2 also need editing for example “The glial cluster (cluster 11), by contrast…” should be just Fig 2D I believe.

The single-cell RNAseq analysis uses a novel method to disaggregate cells that I believe has not appeared previously in the literature (rather than only CMFB+Collagenase dissociation followed by FACS). What was the reason behind this choice of method, are glia cells too large to FACS sort or this papain+Neurobasal better at preserving live glial cells, or perhaps some other reason?

For the future, and out of curiosity, I wondered if glial-absent animals might undergo excessive darkening after sensory overstimulation, which might point to activity dependent accumulation of the pigment. Also I wondered whether these pigments may be related to known pigmentation pathways in planarians, potentially those involving serotonergic precursor substrates. Perhaps it would be possible to genetically ablate such pigment and have a restoration of behavioral activity following overstimulation, which would argue that the pigmentation from glial dysfunction can have a pathological role in this (or other) systems.

Reviewer #3: PGENETICS-D-24-00869

In this manuscript, Scimone and colleagues define a molecular role for Notch signaling in the coupled regeneration of neurons and glia in planarians. This paper addresses an exciting topic and provides a first glimpse into the function of Notch signaling during whole-body regeneration in planarians. It is an excellent fit for PLOS Genetics and I have two major and a few minor suggestions for the authors to consider.

Major:

1. The major critique that I have is that while delta-2+ neurons are very often proximal to notch-1+ glia, no evidence has been presented to show that delta-2+ neurons contact or are in proximity to glial progenitors themselves (as shown in the model figure, Fig. 4C). This type of evidence would help confirm that Notch signaling is critical for glial specification from progenitors and not for local terminal differentiation or transdifferentiation of glia from a more general phagocytic cell type.

a. ISH showing phagocytic progenitors proximal to delta-2+ neurons would markedly strengthen the manuscript and go a long way to support the model presented by the authors.

b. Eye transplantation from lethally irradiated donors into irradiated hosts could confirm that glial engrafting doesn’t occur through a non-stem cell driven mechanism (like transdifferentiation).

c. The expected result if Notch signaling is important for terminal differentiation rather than specification would be cells stalled at an immature state after pathway perturbation. This might be similar to what authors report in Supp. Fig. 1E, where there seem to be a number of remaining estrella- cells marked by the glial pool after notch-1(RNAi) or delta-2(RNAi). What are these cells? What is the % loss of pool+ cells using this marker after RNAi? Where are the remaining cells expressing markers of this pool in the clustering data? Is it possible that another phagocytic cluster actually represents immature glia or something capable of becoming glia?

2. Sarah Elliot wrote a thesis that included data on components of Notch signaling in planarians which is available online at Open Access Theses and Dissertations. Though this is not peer reviewed, I would encourage the authors to consider mentioning and citing this thesis as a prior characterization of the Notch pathway in the main text. In a related point, because the components of this pathway have not been detailed in a peer reviewed paper, I would encourage the authors to include Supp. 1A and a pathway diagram/drawing in the main figures. Expression patterns of these genes, if available, would also be useful. This will provide a starting point for other planarian researchers wanting to dive into this pathway and would improve the impact of this manuscript. I would also be interested in knowing whether the nomenclature in this manuscript matches the nomenclature of genes in the Elliot thesis or whether the numbering of Notch/Delta homologs might differ. Clarity on this point might be warranted so that readers can integrate this manuscript with prior work.

Minor:

1. notch-1 is also expressed in the protonephridia (Fig. 3A). Are glia present in this location as well? If not, how do the authors imagine that glial production near these structures is avoided?

2. One challenge in interpreting Fig. 2C-E is the lack of information about phagocytic cell clusters. To which clusters do categorized cell types like pigment cells belong? How do these clusters relate to previous published work on cell types in this category? Even sharing markers of each subcluster would be helpful when the cluster is uncharacterized.

3. A more rigorous quantification of changes in non-glial cathepsin+ cell types (via ISH in Fig. 1C and in Fig. 2D, with statistical analyses) seems advisable. Numbers of cells in subclusters 1, 3, and 6 also seem decreased in delta-2(RNAi) animals, but I can’t tell how meaningful this is without statistical testing.

4. The pigmentation observed after loss of glia is interesting. Were there any changes in numbers of pigmented cells (which also belong to the phagocytic cluster) or any changes in pigment-associated transcripts? I’m inclined to think this is pigment accumulation in neurons, but it would be helpful to mention evidence that suggests against other possibilities.

5. It would be nice for the authors to comment on why they believe su(H)RNAi animals experience more marked behavioral impacts. The statistical tests used for the behavioral analysis also seem to mark differences for each sample from time zero to each subsequent time point, while a test to analyze differences between RNAi conditions would be more useful.

6. Some methods are not very well documented, reducing reproducibility (e.g. TUNEL).

7. The labeling of many figures (e.g. Fig. 2C, 2E key, 2G asterisks and P ranges, Supp. 2C) is incredibly small. This reader would be grateful for labeling that could be read on a printed page or screen without magnification.

**Have all data underlying the figures and results presented in the manuscript been provided?**

Reviewer #1: **No: **The single cell sequencing data from this paper is on a public database but it is set to not be released until September 2026. This prohibits the public to access and evaluate the data.

Reviewer #2: Yes

Reviewer #3: Yes

PLOS authors have the option to publish the peer review history of their article (what does this mean?). If published, this will include your full peer review and any attached files.

Reviewer #1: No

Reviewer #2: No

Reviewer #3: No

---

## [Decision Letter · Decision Letter 1]

15 Jan 2025

Dear Dr Reddien,

We are pleased to inform you that your manuscript entitled "Coordinated neuron-glia regeneration through Notch signaling in planarians" has been editorially accepted for publication in PLOS Genetics. Congratulations!

Yours sincerely,

Giovanni Bosco, Ph.D.

Section Editor

PLOS Genetics

Giovanni Bosco

Section Editor

PLOS Genetics

Aimée Dudley

Editor-in-Chief

PLOS Genetics

Anne Goriely

Editor-in-Chief

PLOS Genetics

Comments from the reviewers (if applicable):

Reviewer's Responses to Questions

**Comments to the Authors:**

Reviewer #1: I am satisfied with the changes made to this manuscript.

Reviewer #2: This revised version nicely addresses all of my comments, and I congratulate the authors on a fantastic study.

Reviewer #3: I am very satisfied with the changes that the authors made in the manuscript for this review.

It is an excellent paper - a joy to read and very exciting for the fields of glial biology and regenerative biology. I appreciate the opportunity to provide feedback and congratulate the authors on their strong work.

**Have all data underlying the figures and results presented in the manuscript been provided?**

Reviewer #1: Yes

Reviewer #2: Yes

Reviewer #3: Yes

PLOS authors have the option to publish the peer review history of their article (what does this mean?). If published, this will include your full peer review and any attached files.

Reviewer #1: No

Reviewer #2: No

Reviewer #3: No

**Data Deposition**

http://datadryad.org/submit?journalID=pgenetics&manu=PGENETICS-D-24-00869R1

**Press Queries**

---

## [Editor Report · Acceptance letter]

22 Jan 2025

PGENETICS-D-24-00869R1 

Coordinated neuron-glia regeneration through Notch signaling in planarians 

Dear Dr Reddien, 

We are pleased to inform you that your manuscript entitled "Coordinated neuron-glia regeneration through Notch signaling in planarians" has been formally accepted for publication in PLOS Genetics! Your manuscript is now with our production department and you will be notified of the publication date in due course.

With kind regards,

Zsofia Freund

PLOS Genetics

On behalf of:
